# Why is leptospirosis hard to avoid for the impoverished? Deconstructing leptospirosis transmission risk and the drivers of knowledge, attitudes, and practices in a disadvantaged community in Salvador, Brazil

**Fabiana Almerinda G. Palma**[1☯], **Federico Costa**[1,2☯]*, **Ricardo Lustosa**[1], **Hammed O. Mogaji**[1], **Daiana Santos de Oliveira**[2], **Fábio Neves Souza**[3], **Mitermayer G. Reis**[2,4,5], **Albert I. Ko**[2,5], **Michael Begon**[6☯], **Hussein Khalil**[7☯]

1 Federal University of Bahia, Salvador/Institute Health Collective, Salvador, Bahia, Brazil, 2 Instituto Gonçalo Moniz, Fundação Oswaldo Cruz, Ministério da Saúde, Salvador, Bahia, Brazil, 3 Federal University of Bahia/Institute of Biology, Salvador, Bahia, Brazil, 4 Faculdade de Medicina da Bahia, Federal University of Bahia, Salvador, Brazil, 5 Department of Epidemiology of Microbial Diseases, Yale School of Public Health, New Haven, Connecticut, United States of America, 6 Department of Evolution, Ecology and Behaviour, The University of Liverpool, United Kingdom, 7 Department of Wildlife, Fish, and Environmental Studies, Swedish University of Agricultural Sciences, Umeå, Sweden

☯ These authors contributed equally to this work.
* fcosta2001@gmail.com

**Data Availability Statement:** The datasets used and/or analyzed during the current study cannot be

## Abstract

Several studies have identified socioeconomic and environmental risk factors for infectious disease, but the relationship between these and knowledge, attitudes, and practices (KAP), and more importantly their web of effects on individual infection risk, have not previously been evaluated. We conducted a cross-sectional KAP survey in an urban disadvantaged community in Salvador, Brazil, leveraging on simultaneously collected fine-scale environmental and epidemiological data on leptospirosis transmission. Residents' knowledge influenced their attitudes which influenced their practices. However, different KAP variables were driven by different socioeconomic and environmental factors; and while improved KAP variables reduced risk, there were additional effects of socioeconomic and environmental factors on risk. For example, males and those of lower socioeconomic status were at greater risk, but once we controlled for KAP, male gender and lower socioeconomic status themselves were not direct drivers of seropositivity. Employment was linked to better knowledge and a less contaminated environment, and hence lower risk, but being employed was independently associated with a higher, not lower risk of leptospirosis transmission, suggesting travel to work as a high risk activity. Our results show how such complex webs of influence can be disentangled. They indicate that public health messaging and interventions should take into account this complexity and prioritize factors that limit exposure and support appropriate prevention practices.

shared publicly because of personal information of participants in the KAP survey, sero-survey, at the individual and household and peri-domiciliary level. Researchers who wish to access the data can contact the data manager at the Oswaldo Cruz Foundation, Nivison Junior (nivisonjr@gmail.com).

**Funding:** The study was funded by Medical Research Council (UK) (MR/ P024084/1) to MB, Fundação de Amparo à Pesquisa do Estado da Bahia (BR) (1502/2008). The funders had no role in the design, data collection, analysis, decision to publish, or preparation of manuscript.

**Competing interests:** The authors have declared that no competing interests exist.

## Introduction

The incidence of infectious diseases in disadvantaged communities, especially in large urban centers, is higher than incidence elsewhere [1]. There are demographic, socioeconomic, and environmental risk factors contributing to the maintenance of such disparity [2]. Indeed, there is a consensus regarding the importance of socioeconomic differences in explaining ill health [3], and differences in education, employment, and income are consistently associated with health inequalities [4–6]. In disadvantaged neighborhoods, the inadequate environmental and infrastructural conditions, such as the presence of trash, open sewers, and standing water, also contribute to the proliferation of pathogens and their hosts, and increase resident exposure to a range of pathogens [7,8].

In addition to contextual risk factors that directly or indirectly expose people to pathogens, differences in knowledge, attitudes, and practices (KAP) towards diseases also contribute to health disparities. Many recent studies highlight the importance of KAP for COVID-19 [9], tuberculosis [10], malaria [11], leptospirosis [12], and arboviruses such as zika [13] and dengue [14]. Most of these studies rely on the assumption that the risk of acquiring infection is determined by individual patterns of behavior (practices). These practices are assumed to be driven by attitudes, which are determined, in turn, by the level of knowledge of a disease and its associated risk factors [10–14]. However, rarely has KAP been related to objective measures of risk such as probabilities of infection computed from field data [15,16]. While a number of studies have considered the role of demographic and socioeconomic factors in shaping KAP, the role of environmental conditions has been largely overlooked. Furthermore, available KAP studies have implicitly assumed that knowledge, attitudes, and practices all share the same determinants. Indeed, there is an apparent absence of studies that have disentangled the direct effects of socioeconomic [15,16], and environmental variables [17], on disease risk, and also their indirect effects, acting separately, through knowledge, attitudes and practices in urban communities. Strategies to reduce disease risk are often top-down and overlook local knowledge, and more importantly, local barriers, facilitators of preventative measures, and risk perception [2]. Hence, to identify appropriate intervention targets that are more consistent with the reality of the most impoverished populations, it is crucial to trace and quantify the relationships between perceptions and practices of residents, measured through KAP, and the socioeconomic, demographic, and environmental factors as both groups of factors combine, ultimately, to determine risk.

In this study, we present an analysis aimed at disentangling these various and potentially interrelated pathways to leptospirosis transmission risk in the city of Salvador, Brazil. Leptospirosis is a globally distributed and neglected disease that disproportionately affects vulnerable urban residents in low-income communities. Earlier, we have identified socioeconomic and environmental risk factors for leptospirosis [7,18,19], and found a strong link between socioeconomic status, infrastructure and sanitation on the one hand, and exposure to *Leptospira* on the other. Here, we conducted a cross-sectional and detailed KAP study in an urban low-income community, leveraging simultaneously collected fine-scale environmental and epidemiological data, to understand how demographic, socioeconomic, and environmental factors interact to determine KAP of residents, and how the combination of aforementioned factors determines infection risk.

## Materials and Methods

### Study area

Our project, initiated in 2017, is based in four low-income communities in the city of Salvador, Brazil (ca. 2.872.347 million inhabitants in 2019) [20]. In those four communities, we

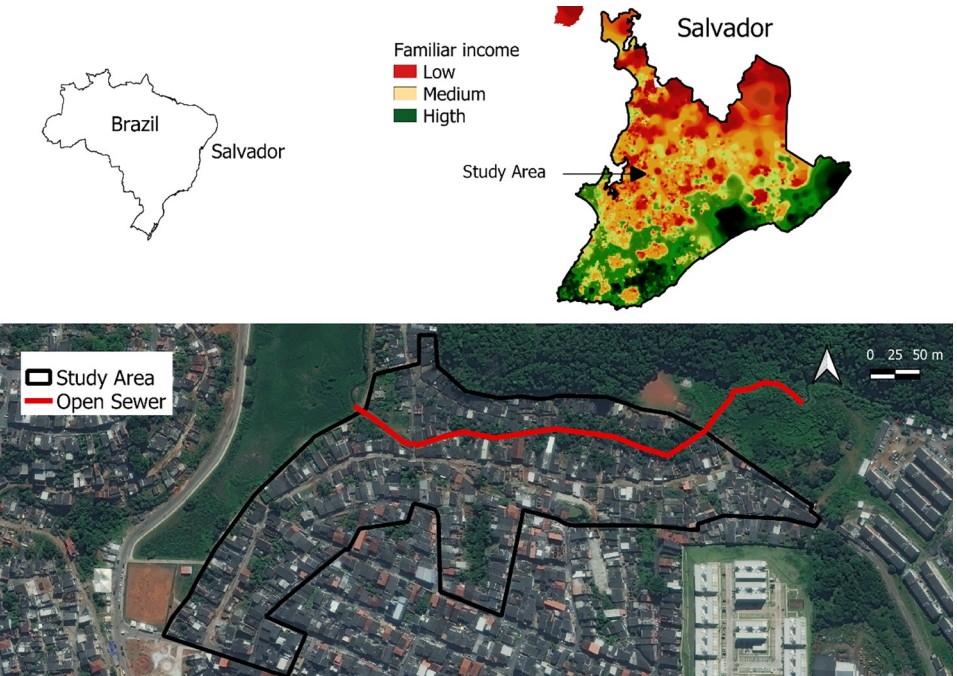

**Fig 1. Socioeconomic and environmental profile of the study area, Marechal Rondon, Salvador, Bahia, Brazil.** The maps are not copyrighted as they were created by the authors using QGIS 2.18 software. The boundary polygons of Brazil, Salvador and the production of the Salvador income map were downloaded from open and publicly accessible base of IBGE—Instituto Brasileiro de Geografia e Estatistica, on the Geosciences platform which can be accessed at https://www.ibge.gov.br/geociencias/downloads-geociencias.html. The WorldView-3 May 2017 satellite image with 31cm resolution was also used to digitize the study area and sewer line. The image was acquired by the research project / Instituto Gonçalo de Moniz—IGM—Fiocruz Bahia from the company Satmap, with disclosure permitted referencing the Copyrights of DigitalGlobe images.

conducted a sero-survey of 1,318 residents in 2018, where trained phlebotomists and nurses collected blood samples and applied individual and household-level questionnaires to document sociodemographic and environmental (household and peri-domiciliary) factors [7].

In this study, we conducted an in-depth KAP survey in one of those four communities, namely Marechal Rondon (MR), located in the northwestern periphery of Salvador (Fig 1). MR is considered a low-income urban community with poor social and environmental conditions (infrastructure and sanitation), as well with high concentration of families in vulnerable areas at risk [21]. Unadjusted seroprevalence for *Leptospira* in MR was 12.0% (n = 338) [7].

First, we identified all residents that had participated in the serological survey (*N* = 338) and, approached those eligible to participate in this KAP study (*N* = 304). We confirmed the adequacy of sample size for this study based on the previously reported 12% seroprevalence for leptospirosis with 95% CI and an assumed non-response rate of 10% [22,23]. Eligible residents were men and women 15 years or more of age, who slept in the household for at least three nights per week, and had participated in the sero-survey in 2018. Trained team members visited the residents between April and June 2019, explained the objectives of the research, before inviting them to participate. Upon recruitment, the team applied a standardized questionnaire to document KAP of the study participants giving the participants a sufficient time to recall. Participants <18 years of age provided written informed assent and their legal guardians provided written informed consent.

## Data collection

The items in our questionnaire were based on previous KAP studies on leptospirosis [12,24,25], and adapted to the context and objectives of this study. The 67 items (S1 Appendix) were organized into four main categories: (1) sources of information on leptospirosis, (2) variables related to knowledge (what leptospirosis is; transmission, signs and symptoms; complications; control and prevention practices), (3) attitudes (individual, household and peri-domiciliary) and (4) practices (individual, household and peri-domiciliary). Demographic, socioeconomic, and environmental data at household and peri-domiciliary level were available from our previous study [7].

For the "knowledge" section, only residents who had heard of leptospirosis proceeded to answer the rest of the questions; those who had never heard of leptospirosis were not considered in subsequent analyses. For answers given by participants to questions based on statements that were true, each "correct" answer was assigned a score of 2, a "do not know" answer was assigned 1, and an "incorrect" answer was assigned 0; for false statements the scales were reversed [25].

For questions on attitudes and practices, we used a three-point Likert scale [25,26]. For each question on satisfactory attitudes, a "yes" answer was given a score of 2, "maybe" a score of 1, and "no" was given 0 score. For preventative (satisfactory) practices, a "yes" answer was given a score of 2, a "sometimes" answer received a score of 1, and a "no" answer was given a 0. For questions on unsatisfactory attitudes and practices, the scales were reversed. We summed the scores for each individual to obtain the total score for each dimension of KAP. Questions on attitudes were related, among others, to the perceived seriousness of the disease, perceived vulnerability to infection, and importance of protecting oneself (for full list see S3 Table and S1 Appendix). For the practices dimension, the questionnaire sought information on protective measures taken by residents to avoid contact with sources of environmental contamination (e.g., wearing gloves when handling trash) and to reduce resources for rats within or near the household (e.g., adequate trash disposal, food storage, etc.). The maximum score for knowledge, attitudes, and practices sections was 60, 30, and 28, respectively.

## Statistical analysis

For all participants, scores for each dimension of KAP were analyzed as the number of points obtained as a proportion of the maximum possible score [12,26]. We combined KAP data with previously collected data on serostatus for leptospirosis, demographic, socioeconomic, and environmental/exposure factors [7,20]. Here, demographic data consisted of the following variables: "gender (male/female)", "age (below/above 40)", and "race (black or mixed/non-black)". Socioeconomic variables were "finished primary school (yes/no)", "family income (lower/higher than minimum wage)", "employment status (employed/ unemployed")", and whether a participant "knows someone who had leptospirosis (yes/no)". Family income was categorized based on whether it was above or below the minimum wage of R$ 950, as defined by the Brazilian Institute of Geography and Statistics [20]. Data on race was also based on IBGE data, which recognized five different races (white, black, mixed, indigenous, Asian), but classified here as either "black" or "non-black" as the majority of residents were black or mixed race (Table 1). The environmental/exposure variables were "Owns animals (yes/no)", "Owns backyard (yes/no)", "Presence of trash within 10 meters of house (yes/no)", "Presence of open sewer within 10 meters of the house (yes/no)", "Walked barefoot in the last 12 months (yes/no)", and "Had contact with sewage in the last 12 months (yes/no)" (See complete dataset attached as S1 File).

**Table 1. Sociodemographic and environmental characteristics of study population in Marechal Rondon, Salvador-Bahia, Brazil (n = 248).**

| Characteristic | Frequency or Median | %, SD or IQR[a] |
|---|---|---|
| **Demographics** | | |
| Male gender | 91 | 36.7 |
| Age (in years) [a] | 44.2 | 18.4 |
| **Socioeconomic indicators** | | |
| Incomplete primary school | 68 | 27.4 |
| Black or mixed race | 226 | 91.1 |
| Household per capita income (in Brazilian Reais) [a] | 300 | 77–485 |
| Occupation (n = 86) | | |
| Informal work [b] | 65 | 75.6 |
| Work-related exposures (n = 86) | | |
| Contact with contaminated environment [c] | 64 | 74.4 |
| Risk occupation [d] | 22 | 25.6 |
| **Individual exposure factors** | | |
| Sighting of rat (last year > 10 meters of house) [e] | 138 | 61.6 |
| Walked barefoot (last year) | 72 | 29.0 |
| Contact with sewage water (last year) | 72 | 29.0 |
| Contact with mud (last year) | 115 | 46.4 |

[a] Standard deviation (SD) (for means) and Interquartile ranges (for medians) are shown for continuous variables of age and per capita household.

[b] Defined by income-generating activities and without a formal contract.

[c] Reported exposure to mud, refuse, flooding water or sewage water in the workplace.

[d] Occupation as construction worker, refuse collector or mechanic, which is associated with a workplace environment characterized by high rat infestation.

[e] Data is missing for twenty-four respondents.

For each dimension of KAP and for seropositivity, we first fitted bivariate analyses using a generalized linear model (GLM) with a binomially distributed error structure. KAP scores and seropositivity were the response variables, and each of the demographic, socioeconomic, and environmental factors were included as predictors. Predictors with p-value < 0.15 were then included in the full model for each dimension of KAP. In addition to these variables, we included knowledge score as a predictor for the attitudes model, and both knowledge and attitudes scores as predictors for the practices model. Finally, in addition to three models (one for each dimension of KAP), we fitted an infection model (GLM with a binomially distributed error structure), with serostatus as the response variable and KAP as predictors, in addition to demographic, socioeconomic, and environmental variables with p-value <0.15 in the bivariate analysis for seropositivity.

For all multivariate models, we used a mixed (forward and backward) selection approach using the *dredge()* function in the statistical software R. To select the final model parsimoniously in each case, we identified the model with the lowest value of Akaike Information Criteria (AIC). If two or further models were within two AIC values of this minimum, we selected the model with the fewest number of predictors. For the final GLM models, we tested for multicollinearity among the predictors using the variance inflation factor (VIF), where a VIF < 10 was considered acceptable [27]. We performed all analyses in R [28], using MuMin() and lme4 () packages [29,30] (See complete script attached as S2 File).

### Ethics statement

This study was approved by the Research Ethics Committee of the Institute of Collective Health/ Federal University of Bahia (CEP/ISC/UFBA) with CAAE number 68887417.9.0000.5030, and a National Research Ethics Committee (CONEP) linked to Brazilian Ministry of the Health under approval numbers 2.245.914–2.245.914.17–3.315.568.

## Results

### Demographic, socioeconomic, and environmental characteristics of study participants

Out of 304 eligible residents, 248 (81.6%) participated in this study (2.3% had moved out of the study site, 0.3% died, 4.6% refused to participate, and 11.2% were not found at home on three or more separate visits). The majority of participants were female (n = 157, 63.3%), with a mean age of 44.2 (range 15–102 years; see Table 1 for a summary of the demographic and environmental variables).

### Knowledge of study participants about leptospirosis

Most participants (92.2%) had previously heard about leptospirosis. Among those, 48.4% identified television/radio as the main sources of information, and 24.0% and 10.2% reported that they obtained the information from neighbors and school, respectively (S1 Table). The majority of participants who had heard about leptospirosis identified it as a disease associated with rats (93.9%) and contact with rat urine, with walking barefoot (98.4%), contact with trash (95.5%) and flood water (93.9%) as the main modes of transmission. However, 22.4% answered that leptospirosis is transmitted *via* mosquito bites and 12.6% believed it can be spread from person to person (S1 Table).

### Attitude and practices of study participants about leptospirosis

Most of the participants viewed leptospirosis a serious disease (98.4%) and had worries about being infected (95.2%). However, 16.9% were not worried about the presence of rats near their household, and 19% responded that they would not participate in leptospirosis prevention and control activities organized by local health centers (S2 Table). Participant responses suggested generally satisfactory preventative practices; 91.9% answered that they avoid contact with sources of contamination. However, 21.0% of the participants reported that they do not use gloves or boots when being in contact with garbage or sewage water, and 49.6% used illegal rodenticides to control rats in their homes (S3 Table).

### Demographic, socioeconomic, and environmental drivers of the knowledge, attitudes, and practices

Knowledge score was associated with demographic, socioeconomic, and environmental factors. Knowledge scores were lower for males compared to females (OR = 0.83, 95% Cl = 0.76–0.92), and for participants who did not finish primary education compared to those who did (OR = 0.73, 95% Cl = 0.66–0.80) (Table 2). Knowledge scores were higher for participants who were employed (OR = 1.13, 95%Cl = 1.02–1.24), who lived in households with a family income > 950 R$ (OR = 1.14, 95% Cl = 1.04–1.24), and participants who personally knew someone who had leptospirosis (OR = 1.23, 95%Cl = 1.12–1.35). Among the environmental variables, having an unpaved backyard was associated with a lower knowledge score compared to those who did not (OR = 0.86, 95% Cl = 0.79–0.95) (Table 2).

**Table 2. Final three multivariate binomial regression (GLM) models explaining knowledge score, attitudes score, and practices score using demographic, socioeconomic, and environmental variables (n = 246).**

| Variables | aOR (95% CI) [a] | P-value [b] |
|---|---|---|
| **Knowledge Score** | | |
| Male | 0.83 (0.76–0.92) | <0.001 |
| Did not finish primary education | 0.73 (0.66–0.80) | <0.001 |
| Employed: yes | 1.13 (1.02–1.24) | 0.018 |
| Family income > R$ 950 (minimum wage) | 1.14 (1.04–1.24) | 0.006 |
| Knows someone who had leptospirosis | 1.23 (1.12–1.35) | <0.001 |
| Has a backyard | 0.86 (0.79–0.95) | 0.002 |
| **Attitude Score** | | |
| Knowledge score | 1.03(1.02–1.05) | <0.001 |
| Male | 0.77 (0.64–0.91) | <0.001 |
| Older than 40 | 1.36 (1.14–1.63) | 0.018 |
| Employed: yes | 1.36 (1.13–1.63) | 0.006 |
| Did not finish primary education | 0.80 (0.65–0.98) | <0.001 |
| Race Black | 1.72(1.35–2.18) | 0.002 |
| Owns animals | 0.71 (0.60–0.84) | <0.001 |
| Presence of trash within 10 meters of house | 1.23 (1.05–1.45) | 0.011 |
| **Practice Score** | | |
| Attitude score | 1.11 (1.09–1.13) | <0.001 |
| Walked barefoot outside | 0.81 (0.67–0.97) | 0.020 |
| Presence of trash within 10 meters of house | 0.82 (0.69–0.98) | 0.028 |
| Contact with sewage in last 12 months | 0.84 (0.70–1.01) | 0.057 |

[a] aOR = Adjusted odds ratio.

[b] p ≤ 0.05.

For attitudes, the final model included knowledge score as a predictor, whereby a higher knowledge score was associated with higher, hence more satisfactory, attitudes score (OR = 1.03, 95% CI = 1.02–1.05) (Table 2). Similar to the knowledge score model, demographic, socioeconomic, and environmental factors were also associated with attitudes score beyond their indirect effects, acting through knowledge. Male participants were more likely to have a lower attitudes score compared to female participants (OR = 0.77, 95% Cl = 0.64–0.91), and those >40 years of age were more likely to have a higher attitudes score compared to younger participants (OR = 1.36, 95% Cl = 1.14–1.63). Being employed was associated with a higher attitude score (OR = 1.36, 95% Cl = 1.13–1.63), and participants who did not finish primary education had lower attitudes scores compared to those who did (OR = 0.80, 95% CI = 0.65–0.98). Participants who consider themselves black race had higher attitudes score compared with non-black participants (OR = 1.72, 95%Cl = 1.35–2.18). Among environmental factors, ownership of animals was associated with a lower attitudes score (OR = 0.71, 95% Cl = 0.60–0.84). On the other hand, the presence of trash within 10 meters of household was associated with a higher attitudes score (OR = 1.23, 95% Cl = 1.05–1.45) (Table 2).

For practices, the final model suggested that a higher practices score, and thus more satisfactory practices, was associated with a higher attitudes score (OR = 1.11, 95% Cl = 1.09–1.13), in addition to two environmental (but not demographic or socioeconomic) variables, which had effects beyond their indirect effects, acting through knowledge and attitudes (Table 2). Lower practices scores were found for participants who had walked barefoot in the past 12

months compared to those who did not (OR = 0.81, 95% Cl = 0.67–0.97) and who had contact with sewage in the past 12 months (OR = 0.84, 95% Cl = 0.70–1.01), the latter association being near significant. Participants who reported the presence of trash within 10 m of the household also had lower practices scores than those who did not (OR = 0.82, 95% Cl = 0.69–0.98) (Table 2).

### KAP as a driver for *Leptospira* seropositivity

Among the 248 participants in this study, 29 (11.7%) were seropositive to leptospirosis. Participants' >40 years of age were more likely to be seropositive for leptospirosis compared to younger participants (OR = 3.45, 95% Cl = 1.27–9.35), as were employed participants compared to unemployed ones (OR = 2.45, 95% Cl = 1.03–5.86). Contact with sewage in the past 12 months was also associated with a higher seropositivity (OR = 3.40, 95% CI = 1.43–8.07). Finally, higher scores for knowledge and practices were associated with lower seropositivity (OR = 0.89, 95% CI = 0.82–0.97 for knowledge; OR = 0.82, 95% CI = 0.70–0.96 for practices) (Table 3).

## Discussion

In this cross-sectional study, we evaluated the contributions of knowledge, attitudes and practices (KAP), and their respective determinants, to *Leptospira* seropositivity in a disadvantaged community in the city of Salvador, Brazil. Our final model for seropositivity indicated that residents who had recent contact with sewage were three times more likely to be seropositive compared to those who did not, and employed residents were more than twice as likely to be positive compared to unemployed residents. However, seropositivity was also lower in residents with higher knowledge of the disease and for those who adopted protective practices. Crucially, knowledge and practices themselves were determined by a range of demographic, socioeconomic and environmental factors. These direct and indirect pathways are summarized in Fig 2.

The scores for knowledge, affecting seropositivity indirectly, were lower in males, in those who did not complete their primary education, and who had a back yard. Knowledge scores were higher in those who were employed, had a family income higher than the minimum wage, and who knew someone who had had leptospirosis. The scores for attitude were higher in those with better knowledge, and so each of the factors affecting knowledge affected attitudes indirectly. Attitude scores were also higher in residents over 40, those who were employed, those who identified as black, and those with trash within 10 meters of their house, and were lower in males, in those who failed to finish primary education, and those who

**Table 3. Final multivariate logistic regression (GLM) predicting seropositivity of residents using KAP, demographic, socioeconomic, and environmental variables (n = 246).**

| | aOR [a] (95% CI) | P-value [b] |
|---|---|---|
| **Variables** | | |
| Older than 40 | 3.45 (1.27–9.35) | 0.015 |
| Employment | 2.45 (1.03–5.86) | 0.044 |
| Contact with sewage in last 12 months | 3.40 (1.43–8.07) | 0.006 |
| Knowledge score | 0.89 (0.82–0.97) | 0.006 |
| Practice score | 0.82 (0.70–0.96) | 0.016 |

[a] aOR = Adjusted odds ratio.

[b] p ≤ 0.05.

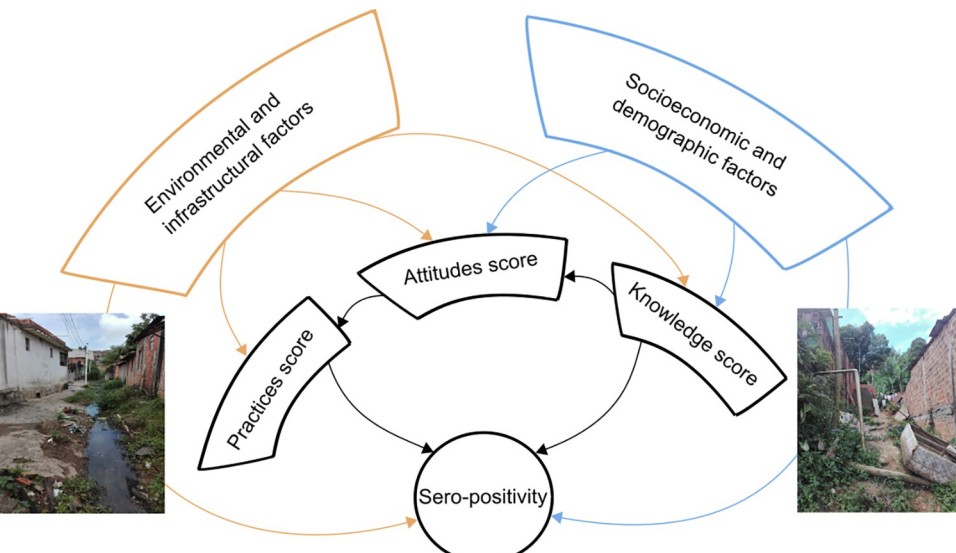

**Fig 2. The network of relationships linking socioeconomic, demographic, and environmental factors to knowledge, attitudes, practices scores, and showing how both groups of factors determine, directly and indirectly, leptospirosis sero-positivity.** The diagram represents the output from four generalized linear models, and the individual variables can be found in Tables 2 and 3. The authors created these figures using the findings generated from the analysis of their primary data.

owned animals. Finally, the factors that drive practices affect exposure indirectly, and so each of the factors affecting attitudes, including those acting indirectly through knowledge, affected practices indirectly. Practices scores were also lower in those who walked barefoot, those with trash within 10 meters of their house, and those who had come into contact with sewage in the last 12 months.

Of the many pathways and drivers of *Leptospira* seropositivity identified, a number of pathways are particularly notable. First, our earlier research has shown that males had a higher risk of leptospirosis [7,18], a pattern commonly observed for several environmentally-transmitted and vector-borne diseases [31–33]. Both behavioral and physiological differences between men and women contribute to sex/gender biases in risk [34]. Also, cultural variables, such as gender stereotypes, contribute to differentials in attitudes to risk [35]. Here, we found that men had lower knowledge and attitudes scores compared to women, which contributed directly and indirectly, through sub-optimal practices, to an observed higher risk of infection in men. However, once we controlled for KAP, male gender was not a significant (direct) driver of seropositivity, suggesting that gender-specific attitudes and exposure patterns [35], and not differences in physiology, were responsible for the higher leptospirosis risk in males observed in previous studies. Indeed, gender biases in leptospirosis risk become even larger in rural settings, due to higher *Leptospira* exposure for men during agricultural and fishing activities [34].

In previous studies, lower scores for socioeconomic variables such as income and education have been linked to a higher risk of leptospirosis and other zoonotic and vector-borne diseases [7,18,19,36]. Here too, higher family income, completing primary education, and being employed were associated with higher knowledge scores, and being employed was also associated with better attitudes. Hence, higher socioeconomic status leads to better knowledge, attitudes, and consequently more satisfactory practices that ultimately contributed to lower seropositivity. These patterns are thus consistent with previous studies, indicating that even after controlling for demographic and environmental factors, lower socioeconomic status was a direct predictor of leptospirosis risk [7]. Notably, however, socioeconomic status here had no

direct negative impact on seropositivity, suggesting KAP as the causal route to such health inequalities.

Furthermore, having employment was independently associated with a higher, not a lower risk of *Leptospira* exposure (Fig 2). In an earlier study, we proposed that exposure away from the household is understudied and remains difficult to account for [7]. Our present results support this view, suggesting that while employment, among other proxies of higher socioeconomic status, are linked to better knowledge and a less contaminated peri-domestic environment [7], work- or travel-related exposure away from the household remains inevitable given the inadequate environmental and sanitary conditions that are ubiquitous in the community. Frequent large-scale flooding events that affect public and private spaces, inadequate infrastructure, and poor access to main roads and public transit make travel to work and work itself risky activities, with negative consequences on health and potentially income [37]. This inability to avoid environmental contamination due to lack of alternatives is a common challenge in low-income settings. In northern Senegal, for example, villagers were aware of schistosomiasis risk in nearby lakes, but had no option but to continue fishing and agricultural activities in or in close proximity to those lakes [38].

Residents who identified as black/mixed race had better attitudes towards leptospirosis compared to non-black residents. However, there were no differences between the two groups in knowledge, practices, or seropositivity, in contrast to previous studies, which showed that black/mixed residents were at a higher risk of leptospirosis [18]. Our findings therefore suggest that while residents of black/mixed ethnicity disproportionately recognized leptospirosis as a potentially serious health issue, they were at a similar (and in previous studies higher) risk of leptospirosis compared to non-black residents. Our findings are in line with those reported for other diseases and contexts. Historically marginalized ethnic groups in the US were more likely to perceive COVID-19 as a major threat to society [39], yet incidence and severe COVID-19 infections are disproportionately higher in marginalized ethnic minorities [40]. Hospitalizations due to infectious diseases in general are higher for indigenous ethnic communities, especially for the most deprived groups [41]. Here, despite the better attitudes observed towards disease risk, socioeconomic and environmental inequalities, manifested by types of employment, living and environmental conditions, and sociodemographic factors, kept marginalized ethnic groups at a disproportionate risk of infectious diseases [42].

Previous studies on leptospirosis have reported that poor environmental and sanitary conditions increase the risk of infection [18,19,24,43]. Here we found an additional, indirect contribution of poor environmental conditions to *Leptospira* risk, acting through KAP. For example, while the presence of trash near the household contributed to better attitudes towards leptospirosis, as it likely made residents aware of high levels of rat infestation, it nevertheless made residents less likely to adopt satisfactory practices. Further, recent exposure to sewage, in addition to directly contributing to higher seropositivity, also made residents less likely to adopt satisfactory practices. Indeed, no socioeconomic or demographic factor directly influenced practices, and thus their effect was strictly indirect, mediated through knowledge and attitudes scores. Inadequate environmental conditions, therefore, appeared to form barriers against the adoption of protective practices. We hypothesize that a degraded peri-domiciliary environment, e.g., the presence of trash piles near the household, despite being associated with better attitudes, results in protective practices being seen as useless or of limited value given the overwhelming physical reality that expose residents to a wide range of risk factors. In future studies, residents should be specifically asked about their degree of belief in the effectiveness of protective measures, which could then be tested against environmental and infrastructural conditions near the household. Indeed, variables representing inadequate environmental conditions were doubly important, as they not only contributed to

seropositivity indirectly by making residents less likely to adopt preventative practices, but also directly, after KAP had been taken into account.

This study has several limitations. The first is the impossibility of establishing causality for the associations presented here given the cross-sectional design of the study. Another limitation is related to memory bias of participants, since the variables related to exposure in the year were self-reported.

## Conclusion

Our results indicate that to understand the links between KAP and disease risk, we need to account for the pathways by which demographic, socioeconomic, and environmental factors influence the KAP dimensions and the risk of human leptospirosis, and the options available for residents to mitigate it. Irrespective of factors such as education, income, knowledge, attitudes, and protective measures taken by residents, the inadequacy of the environment and socioeconomic disadvantage remain central, as they render residents helpless in avoiding exposure and ultimately becoming infected. Our findings are important for identifying potential barriers that limits prevention practices and continue to expose residents, and suggest that public health messaging and interventions should take into account both the socioeconomic and environmental determinants of at-risk groups.

## Supporting information

**S1 Checklist. STROBE statement—checklist of items that should be included in reports of *cross-sectional studies.***
(DOC)

**S1 Appendix. List of KAP questions.**
(DOCX)

**S1 Table. Summary of knowledge questions on leptospirosis (n = 246).**
(DOCX)

**S2 Table. Summary of attitudes questions on leptospirosis (n = 248).**
(DOCX)

**S3 Table. Summary of practices questions on leptospirosis (n = 248).**
(DOCX)

**S1 File. Complete dataset.**
(R)

**S2 File. Complete script.**
(TXT)

## Acknowledgments

We deeply appreciate all the residents and community leaders of the neighborhood Marechal Rondon for their contributions in supporting the research and participation in this study. We thank the members of the project for their constructive suggestions for this manuscript.

## Author Contributions

**Conceptualization:** Fabiana Almerinda G. Palma, Federico Costa, Michael Begon, Hussein Khalil.

**Data curation:** Fabiana Almerinda G. Palma, Federico Costa, Daiana Santos de Oliveira, Hussein Khalil.

**Formal analysis:** Fabiana Almerinda G. Palma, Federico Costa, Fábio Neves Souza, Hussein Khalil.

**Funding acquisition:** Federico Costa, Mitermayer G. Reis, Albert I. Ko, Michael Begon.

**Investigation:** Fabiana Almerinda G. Palma, Federico Costa, Daiana Santos de Oliveira, Hussein Khalil.

**Methodology:** Fabiana Almerinda G. Palma, Federico Costa, Hussein Khalil.

**Project administration:** Fabiana Almerinda G. Palma, Federico Costa, Mitermayer G. Reis, Albert I. Ko, Hussein Khalil.

**Resources:** Fabiana Almerinda G. Palma, Federico Costa, Ricardo Lustosa, Daiana Santos de Oliveira, Hussein Khalil.

**Software:** Fábio Neves Souza, Hussein Khalil.

**Supervision:** Federico Costa, Mitermayer G. Reis, Albert I. Ko, Michael Begon, Hussein Khalil.

**Validation:** Federico Costa, Michael Begon, Hussein Khalil.

**Visualization:** Fabiana Almerinda G. Palma, Ricardo Lustosa, Fábio Neves Souza, Hussein Khalil.

**Writing – original draft:** Fabiana Almerinda G. Palma, Federico Costa, Hammed O. Mogaji, Fábio Neves Souza, Michael Begon, Hussein Khalil.

**Writing – review & editing:** Fabiana Almerinda G. Palma, Federico Costa, Hammed O. Mogaji, Fábio Neves Souza, Michael Begon, Hussein Khalil.

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
