## [Decision Letter · Decision Letter 0]

9 Jul 2022

PGPH-D-22-00525

Why is leptospirosis hard to avoid for the impoverished? Deconstructing leptospirosis transmission risk and the drivers of knowledge, attitudes, and practices in a disadvantaged community in Salvador, Brazil

Dear Dr. Costa,

Thank you for submitting your manuscript to PLOS Global Public Health. After careful consideration, we feel that it has merit but does not fully meet PLOS Global Public Health’s publication criteria as it currently stands. Therefore, we invite you to submit a revised version of the manuscript that addresses the points raised during the review process.

We look forward to receiving your revised manuscript.

Kind regards,

Syed Shahid Abbas, MBBS, MPH, Ph.D.

Academic Editor

Journal Requirements:

1. Please update your online Competing Interests statement. If you have no competing interests to declare, please state: “The authors have declared that no competing interests exist.”

2. Please include copy of Tables 5 and 6 which you refer to in your text on page 16. Or if the tables are no longer to be included as part of the submission please remove all reference to it within the text.

Additional Editor Comments (if provided):

Reviewers' comments:

Reviewer's Responses to Questions

**Comments to the Author**

1. Does this manuscript meet PLOS Global Public Health’s publication criteria? Is the manuscript technically sound, and do the data support the conclusions? The manuscript must describe methodologically and ethically rigorous research with conclusions that are appropriately drawn based on the data presented.

Reviewer #1: Yes

Reviewer #2: Yes

2. Has the statistical analysis been performed appropriately and rigorously?

Reviewer #1: Yes

Reviewer #2: Yes

3. Have the authors made all data underlying the findings in their manuscript fully available (please refer to the Data Availability Statement at the start of the manuscript PDF file)?

Reviewer #1: No

Reviewer #2: Yes

4. Is the manuscript presented in an intelligible fashion and written in standard English?

Reviewer #1: Yes

Reviewer #2: Yes

5. Review Comments to the Author

Reviewer #1: Review of “Why is leptospirosis hard to avoid for the impoverished? Deconstructing leptospirosis transmission risk and the drivers of knowledge, attitudes, and practices in a disadvantaged community in Salvador, Brazil.”

This is an important study mapping how knowledge and understanding of leptospirosis shape individual health outcomes in an urban, disadvantaged community in Salvador Brazil. Using survey data, the study shows a relationship between these and knowledge, attitudes, and practices (KAP), and more importantly, their web of effects on individual infection risk conditional on socioeconomic and environmental risk factors for leptospirosis. In my opinion, this manuscript has potential and should be considered for publication. I would suggest the following revisions to improve the manuscript’s contributions.

The survey uses answers to questions that were restricted to 3 Likert scale options. This 3-option format has tradeoffs for measurement. The authors could discuss these limitations. For example, one of the limitations is that individuals with advanced and a more limited understanding are pooled together. There are other tradeoffs and these, and it would be essential to alert and recognize these tradeoffs.

The authors should justify the models they chose for hypothesis testing. Why a GLM with a binomially distributed error structure?

Since the authors argue that there are complex determinants to KAP, I am less convinced that the results of the fitted bivariate analyses for each dimension of KAP and for seropositivity using a generalized linear model (GLM) with a binomially distributed error structure deserve as much discussion in the study.

Instead, I would encourage the authors to devote more attention in the manuscript to reporting and discussing the increases in the odds-ratios holding constant specific demographic, socioeconomic, and environmental characteristics in the multivariate models. In other words, I encourage the authors to more fully develop the analyses reported in Tables 2 and 3, and perhaps add 2 figures.

Smaller comments:

Line 170 “we used a mixed (forward and backward) selection approach using the dredge() function in the statistical software R.” Please explain this choice and justify. Why does this matter (reduce bias, increase efficiency, etc)?

The authors state that data will be available upon request. As a reviewer, I would like to access the data and code.

The abstract needs to be tightened, and the policy implications made more clear.

I encourage the authors to revise the results in line with my recommendations above. In other words, please make sure to acknowledge that marginal effects are based on XX assumptions about the other characteristics in the model, etc.

Reviewer #2: Manuscript : Why is leptospirosis hard to avoid for the impoverished? Deconstructing leptospirosis transmission risk and the drivers of knowledge, attitudes, and practices in a disadvantaged community in Salvador, Brazil (PGPH-D-22-00525) submitted in PLOS Global public health

The manuscript can be accepted for publication after the incorporation of following minor comments

1. In method sections how's authors calculate samples size? which is one of most part of KAP survey

6. PLOS authors have the option to publish the peer review history of their article (what does this mean?). If published, this will include your full peer review and any attached files.

**Do you want your identity to be public for this peer review?** For information about this choice, including consent withdrawal, please see our Privacy Policy.

Reviewer #1: No

Reviewer #2: No

---

## [Decision Letter · Decision Letter 1]

11 Nov 2022

Why is leptospirosis hard to avoid for the impoverished? Deconstructing leptospirosis transmission risk and the drivers of knowledge, attitudes, and practices in a disadvantaged community in Salvador, Brazil

PGPH-D-22-00525R1

Dear Prof Costa,

We are pleased to inform you that your manuscript 'Why is leptospirosis hard to avoid for the impoverished? Deconstructing leptospirosis transmission risk and the drivers of knowledge, attitudes, and practices in a disadvantaged community in Salvador, Brazil' has been provisionally accepted for publication in PLOS Global Public Health.

Best regards,

Syed Shahid Abbas, MBBS, MPH, Ph.D.

Academic Editor

Reviewer's Responses to Questions

**Comments to the Author**

1. If the authors have adequately addressed your comments raised in a previous round of review and you feel that this manuscript is now acceptable for publication, you may indicate that here to bypass the “Comments to the Author” section, enter your conflict of interest statement in the “Confidential to Editor” section, and submit your "Accept" recommendation.

Reviewer #1: All comments have been addressed

2. Does this manuscript meet PLOS Global Public Health’s publication criteria? Is the manuscript technically sound, and do the data support the conclusions? The manuscript must describe methodologically and ethically rigorous research with conclusions that are appropriately drawn based on the data presented.

Reviewer #1: Yes

3. Has the statistical analysis been performed appropriately and rigorously?

Reviewer #1: Yes

4. Have the authors made all data underlying the findings in their manuscript fully available (please refer to the Data Availability Statement at the start of the manuscript PDF file)?

Reviewer #1: Yes

5. Is the manuscript presented in an intelligible fashion and written in standard English?

Reviewer #1: Yes

6. Review Comments to the Author

Reviewer #1: (No Response)

7. PLOS authors have the option to publish the peer review history of their article (what does this mean?). If published, this will include your full peer review and any attached files.

**Do you want your identity to be public for this peer review?** For information about this choice, including consent withdrawal, please see our Privacy Policy.

Reviewer #1: No
